# Development of Optimized Feed for Lipid Gain in *Zophobas morio* (Coleoptera: Tenebrionidae) Larvae

**DOI:** 10.3390/ani13121958

**Published:** 2023-06-12

**Authors:** Tae-Won Goo, Dooseon Hwang, Kyu-Shik Lee, Seung Hun Lee, Eun-Young Yun

**Affiliations:** 1Department of Biochemistry, College of Medicine, Dongguk University, Gyeongju 38766, Republic of Korea; gootw@dongguk.ac.kr; 2Department of Integrative Biological Sciences and Industry, Sejong University, Seoul 05006, Republic of Korea; h.michael8837@gmail.com (D.H.); g-d-lsh@hanmail.net (S.H.L.); 3Department of Pharmacology, College of Medicine, Dongguk University, Gyeongju 38766, Republic of Korea; there1@dongguk.ac.kr

**Keywords:** *Zophobas morio* larvae, lipid gain feed, food waste

## Abstract

**Simple Summary:**

The aim of this study was to develop feed using organic waste to increase the lipid content of the super mealworm *Zophobas morio* larvae (ZML). Feed prepared from fermented food waste (FFW) using three microorganism species was found to be suitable for increasing the lipid content of ZML. These results revealed that feed made from FFW is an environmentally friendly and economical feed suitable for breeding ZML for biodiesel production.

**Abstract:**

Super mealworm *Zophobas morio* (Coleoptera: Tenbrionidea) larvae (ZML) are being investigated as potential candidates for biodiesel production. Several studies have revealed that the crude fat content of ZML can be enhanced by increasing the feed consumed. We aimed to develop an optimized ZML feed that enhances the lipid gain using 10 different ingredients. The results revealed that the highest lipid content was observed in ZML fed food waste (FW). Furthermore, we found that the weight gain of ZML improved when fed fermented FW using three selected microorganisms (3M), *Lactobacillus fermentum*, *Lactobacillus acidophilus*, and *Pediococcus acidilactici*. We also analyzed the effects of preservatives on the weight gain of ZML, and the results revealed that ZML fed 5-day 3M-fermented FW (FFW) containing 0.05% sorbic acid exhibited the highest weight gain. Based on these findings, we produced solid FFW containing 0.05% sorbic acid using 5% agar and established a manufacturing process. Body composition analysis revealed that the lipid content of the ZML fed manufactured feed was higher than that of the ZML fed wheat bran. Therefore, this study suggests that solid FFW containing 0.05% sorbic acid should be used as a commercial feed for ZML breeding to enhance lipid gain, making it an economical substrate for raw biodiesel production.

## 1. Introduction

The demand for eco-friendly and renewable fuels such as biodiesel has been globally increasing because of the unstable supply and price of fossil diesel, and the need to reduce greenhouse gas emissions and environmental pollutants [1]. Biodiesel can be produced from natural lipids found in vegetable oils, animal fats, and waste cooking oil [2]. Edible vegetable oils are the most commonly used raw materials for biodiesel production [3,4]. However, vegetable oil is not an economical option for reducing the production cost of biodiesel because vegetable prices are constantly increasing [2]. Furthermore, many soil resources and vast forests have been destroyed to increase vegetable oil production, leading to other serious environmental problems [5].

Some reports have suggested that non-edible oils, such as *Croton megalocarpus* oil and *Jatropha curcas* oil, are useful for reducing the production costs of biodiesel [6]. However, their long growth duration and environmental impacts, including deforestation and soil resource destruction, are major concerns. Recently, many studies have suggested insects as an alternative, viable, and valuable feedstock for low-cost biodiesel production [2,7,8,9,10]. Insect larvae contain a high crude fat content (approximately 16.7–57.9% of total dried body weight), and their fat levels can be modified by adjusting the feed ingredients [2,11]. Furthermore, insects can degrade organic waste and some plastics; thus, biodiesel production using insects can be an eco-friendly and economic strategy for preventing environmental problems [12,13,14,15,16,17].

The supermealworm, *Zophobas morio* (Coleoptera: Tenbrionidea), and its larvae (ZML) are well-known edible insects used as feed for birds, reptiles, and fish [18,19,20]. Some studies have suggested that it could be used as an alternative feedstock for biodiesel production [7,21]. ZML has also been reported to biodegrade plastics such as low-density polyethylene and polystyrene [22,23]. Investigations have revealed that various organic wastes, including vegetable waste, green garden waste, manure, and crop waste, can be effectively degraded by ZML [23,24]. Dried ZML has been reported to contain 35.0–43.6% crude fat [20,25,26,27,28,29,30,31]. These investigations have demonstrated that ZML can be used as a means of producing biodiesel without environmental problems. Therefore, we attempted to develop an inexpensive feed to improve the lipid gain in ZML and reduce the production cost of biodiesel using ZML.

## 2. Materials and Methods

### 2.1. Materials

We used *Tenebrio molitor* fecal soil (TF), feed-conversed food waste (FW), and agricultural by-products, such as rice bran (RB), wheat bran (WB), corn stalk (CS), defatted sesame meal (SO), defatted perilla meal (PO), soybean residue curd meal (SC), and coffee grounds (CG) as feeds for ZML. RB, CS, defatted soybean meal (SM), SO, PO, and SC were gifts from farms in Gongju (Republic of Korea). CG and FW were obtained from CAFÉ DICTIONARY (Seoul, Republic of Korea) and Huindol Co., Ltd. (Cheonan, Republic of Korea), respectively. WB, TF, and ZML were purchased from Michinmilwum Co. (Yongin, Republic of Korea).

### 2.2. Analyses of the Nutritional Composition of Ingredients and ZML

Composition analyses of the 10 feed ingredients and ZML fed each ingredient was conducted according to the official methods of the Association of Official Analytical Chemists. The crude protein and total nitrogen contents of the ingredients and ZML were analyzed using a FossTecator digestion system (Hilleroed, Denmark) and a Vadopset 50S automatic nitrogen quantitative analyzer (Königswinter, Germany), respectively. Lipid content was analyzed using the ST 243 Soxtec solvent extraction system. To analyze the ash content, 100 g of the sample was transferred to an incinerator, heated at 550 °C to 600 °C until it was completely ash, cooled at room temperature in a desiccator, and weighed. The carbohydrate content was determined as follows: carbohydrate mass (g) = dry matter (100 g) – {crude protein mass (g) + lipid mass (g) + ash mass (g)}. Calories in the ingredients were calculated as 4 Kcal/g for carbohydrates and proteins and 9 Kcal/g for lipids.

### 2.3. Selection of Effective Fermenting Microorganisms

We evaluated the fermentation capacity of microorganisms such as *Lactiplantibacillus plantarum* (KACC 15357), *Limosilactobacillus fermentum* (KACC 11441), *Lactobacillus acidophilus* (KACC 12419), *Pediococcus acidilactici* (KACC 12307), *Bacillus subtilis* (KACC 17047), and *Saccharomyces cerevisiae* (KACC 30008), which were obtained from Korea Agricultural Collection in Rural Development Administration (Wanju, Republic of Korea) and selected as suitable fermentation microorganisms. Effective microorganisms (EM), a fermentation medium, was purchased from Ever Miracle Co., Ltd. (Jeonju, Republic of Korea) and used as a positive control to evaluate the fermentation capacity [32]. *L. plantarum* and *P. acidilactici* were cultured in deMan Rogosa Sharpe (MRS) medium at 30 °C, *L. fermentum* and *L. acidophilus* were cultured in MRS medium at 37 °C, *B. subtilis* was cultivated in nutrient agar medium at 30 °C, and *S. cerevisiae* was cultured in yeast extract peptone dextrose medium at 30 °C for 18 h. The media were then collected by centrifugation at 9000× *g* to evaluate enzyme activities, including amylase, maltase, sucrase, protease, and lipase. Enzyme activity was measured using commercial enzyme assay kits. Amylase and lipase activity assay kits were purchased from Merck KGaA (Darmstadt, Germany), the maltase activity assay kit was obtained from Elabscience Biotechnology (Houston, TX, USA), the EnzyChrom^TM^ Invertase Assay Kit for analyzing sucrase activity was purchased from BioAssay Systems (Hayward, CA, USA), and the Pierce^TM^ Colorimetric Protease Assay Kit was obtained from Thermo Fisher Scientific (Waltham, MA, USA).

### 2.4. Determination of the Optimal Fermentation Period for Producing Feed Using FW

Three microbial strains, *L. fermentum*, *L. acidophilus*, and *P. acidilactici*, were used to ferment FW. The microorganisms were mixed, and 50 mL of the microorganism mixture was added per kg of the FW. FW containing the microorganism mixture was cultured at 27 °C for 5, 7, 14, 21, and 28 days, respectively. After each fermentation period, the FW was transferred to a freezer at −40 °C to stop fermentation. Fermented FW (FFW) obtained after 5, 7, 14, 21, and 28 days of fermentation was used as feed for ZML breeding. To determine the optimal fermentation period for producing feed using FW, we measured the body weight gain of the ZML fed FFW at different time intervals.

### 2.5. Selection of the Preservative Type and Content for Producing Feed Using FFW

Sorbic acid (final concentrations: 0.05%, 0.10%, and 0.15%; Daejung Chemicals and Metals Co., Ltd., Siheung, Republic of Korea) or grapefruit seed extract (GSE; final concentrations: 0.05%, 0.10%, and 0.15%; ES Food Co., Ltd., Gunpo, Republic of Korea) were mixed well with FFW. The mixture was used as a feed for ZML breeding, and the body weight gain of ZML fed with FFW was measured at different time intervals to select the optimal preservative.

### 2.6. Selection of Solidifying Materials for Producing Feed Using FFW

We mixed solidifying materials such as agar, carrageenan, and starch (ES Food Co., Ltd., Pocheon, Republic of Korea) with FFW containing preservatives, each at 20% of the total weight of the mixture. Distilled water was added to the mixture at a ratio of 1:1 (*w*/*v*). The mixture was kneaded using a dough mixer (SAD-2003; Samwoo Industry, Seoul, Republic of Korea), and the dough was extruded using a mincing machine (MN-225; Hankook Fuji Industries, Hwaseong, Republic of Korea) to obtain noodle-like shapes. It was subsequently dried for 50, 100, and 150 min at 60 °C, respectively, and procured as feed for ZML breeding. To select the optimal solidifying material, the body weight gain of the ZML fed with FFW containing a preservative was measured at different time intervals.

### 2.7. Calculation of the Body Weight Gain of ZML

Fifty ZML were bred in 1 kg of each feed, and the total weight of 50 ZML was assessed. ZML were reared for 20 days at 25 °C and 60% relative humidity. The total weight of 50 ZML was measured every 5 days during the rearing period. The average body weight per ZML was determined by dividing the total weight by the number of ZML. The formula used to calculate the body weight gain of ZML was as follows: body weight gain (mg/ind.) = {(measured total weight) − (initial total weight)}/number of ZML. Ind. indicates individual.

### 2.8. Statistical Analyses

Significant differences were determined using a one-way analysis of variance with Duncan’s post-hoc test or Student’s *t*-test using SPSS ver. 18 (SPSS Inc., Chicago, IL, USA). All experiments were performed independently and in triplicates. Each value is presented as the mean ± standard deviation.

## 3. Results

### 3.1. The Weight Gain of ZML Bred with Organic Wastes

We selected 10 types of organic waste as feed for the ZML and analyzed their compositions. We then measured the weight gain of ZML bred with each ingredient over different time intervals. As shown in Table 1, RB and CG had the highest gross energy and crude fat contents. In contrast, Table 2 reveals that SC, CG, TF, and FW contained high levels of carbohydrates. We also found that the highest crude protein content was found in SM. Based on weight gain analyses, the highest weight gain was measured in ZML bred with WB (166.46 ± 43.16 mg/ind.) or FW (159.90 ± 62.85 mg/ind.). Interestingly, despite the differences in nutrient composition, WB and FW were both found to be suitable for increasing the weight gain of ZML. Therefore, these results reveal that WB and FW can be used to improve the production of ZML for biodiesel.

### 3.2. The Composition of ZML Bred with Organic Wastes

We analyzed the compositions of ZML bred with 10 different single ingredients. Our results revealed that ZML bred with FW had the highest CF content, indicating that the lipid content in ZML was enhanced with the use of FW as a feed (Table 3). Although RB and CG had higher CF contents than the other ingredients, the CF content in ZML bred with RB or CG was significantly lower than that in ZML bred with FW. Therefore, we selected FW as the feed to increase the lipid content in ZML.

### 3.3. The Selection of Microbial Strains to Prepare Fermented Feed Using FW

To improve lipid gain using FW and enhance the uptake of the feed, we attempted to ferment FW. For this purpose, we selected six previously reported microbial strains and analyzed their fermentation capacities [33,34,35]. *L. fermentum* exhibited the highest activities of amylase and maltase, whereas *L. acidophilus* exhibited the highest sucrase activity (Table 4). Furthermore, *L. fermentum*, *L. acidophilus,* and *P. acidilactici* exhibited higher protease and lipase activities than other strains (Table 4). To enhance the uptake of proteins and lipids from the feed, it is advantageous to decompose them into low-molecular-weight proteins and lipids. Therefore, we selected three microbial strains (3M), *L. fermentum*, *L. acidophilus*, and *P. acidilactici*, which exhibited high protease and lipase activities, to prepare fermented ZML feed from FW.

### 3.4. Analysis of Weight Gain of ZML Bred with Fermented FW

To investigate whether 3M-fermented FW could improve the growth of ZML, we compared it with EM- or yeast (*S. cerevisiae*)-fermented feed by evaluating the weight gain. As shown in Table 5, the weight gain of ZML fed with 3M-fermented FW for 20 days was higher than that of ZML fed with EM- or yeast-fermented FW. Furthermore, the highest weight gain was observed in ZML fed with 3M-fermented FW for 5 days (Table 5 (C)). Therefore, we selected the 5-day 3M-fermented FW (FFW) as a lipid gain-enhancing feed for ZML because it could shorten production time and reduce production cost.

### 3.5. Determination of Preservatives for FFW

To determine the optimal preservatives for FFW, we analyzed the effect of preservatives on the weight gain of ZML. Sorbic acid and grapefruit seed extract are generally used as feed preservatives. Therefore, we assessed weight gain in ZML fed FFW supplemented with various concentrations of sorbic acid or grapefruit seed extract. The results revealed that the average body weight gain of ZML was higher when fed FFW containing sorbic acid than when fed FFW containing grapefruit seed extract for 20 days (Table 6). The prices of sorbic acid and grapefruit seed extract per 1 kg are approximately ₩ 5.5 and ₩ 9.6 in Korea, respectively. Therefore, we selected 0.05% sorbic acid as a preservative for FFW because it is cheaper than grapefruit seed extract, and the average weight gain of ZML fed FFW containing sorbic acid (FFWS) was higher than that of ZML fed FFW containing grapefruit seed extract. Furthermore, no significant differences were observed in the weight gain of ZML based on the concentration of sorbic acid.

### 3.6. Determination of Solidifying Material for FFWS

To prepare a solid feed using FFWS, we evaluated the weight gain of ZML fed with FFWS with solidifying materials such as agar, carrageenan, or starch. The composition analysis reveals that all diets exhibited a comparable composition (Appendix A). As shown in Table 7, the weight gains of the ZML fed with solidified FFWS were higher than those of ZML fed with WB or FW. Furthermore, ZML fed FFWS containing solidifying materials that were dried for 50 min showed higher weight gain, whereas lower weight gain was observed in ZML fed FFWS containing solidifying materials that were dried for longer periods (100 and 150 min, respectively). Moreover, the highest weight gain was observed in the ZML fed with solid FFWS containing agar dried for 50 min. These results reveal that the solid feed, including the FFWS containing agar as a solidifying material, was suitable for improving the weight gain of ZMLs.

### 3.7. Establishment of Manufacturing Process for the Production of ZML with Lipid Gain and Solid Feed and Analyses of the ZML Composition

We established a manufacturing process for producing a solid feed using FFWS to increase the lipid content of ZML, as presented in Figure 1A. We subsequently analyzed the composition of ZML fed with the manufactured feed and found that the CF content was higher than that of ZML fed with WB, which is commonly used as a specialized feed for mealworms and super mealworms, as shown in Figure 1B. These results revealed that the manufactured feed was more suitable for lipid gain in ZML than WB. 

## 4. Discussion

ZML is a well-known feed and food source used in livestock and human food supplements [20,36,37]. Furthermore, ZML oil is a potent source of biodiesel [7,21]. However, substrate cost is a key issue in breeding ZML for feed, food, and biodiesel. Recently, the price of fossil fuels has risen sharply because of an unstable international situation, and many global companies are now investing in renewable energy. The need to increase biodiesel production to reduce greenhouse gas emissions has been focused on. However, the cost of biodiesel production remains high, and the supply of feedstock has been largely affected by external factors, such as the international situation and climate. Therefore, the production and industrial use of biodiesel are still lower than those of fossil fuels.

In this study, we developed a ZML feed to enhance lipid gain using FW fermented with three different bacterial strains. As shown in Table 3, the lipid contents of ZML fed with FW were the highest, and that of ZML fed with solid FFWS containing agar dried for 50 min was higher than that of ZML fed with WB (Figure 1B). These results confirm that FFW should be used as a feed for ZML to increase lipid contents in the body.

WB is widely used as a feed for ZML breeding. However, the price of WB is gradually increasing owing to its use as a raw material in various food and feed industries. Furthermore, WB supply has been affected by external factors. Therefore, breeding ZML using WB is not an economic strategy for biodiesel production because of the high manufacturing costs. On the other hand, FW not only requires a purchase cost but can also incur disposal costs. In Korea, FW disposal companies pay approximately KRW 16.2 per ton for FW disposal. In contrast, the price of WB in Korea is approximately KRW 150–170 per ton. Therefore, the production cost for solid lipid gain feed using FFWS was found to be much lower than that of WB. Moreover, solid feed production using FFWS is an effective strategy for increasing the recycling rate of FW and preventing the environmental problems caused by FW.

## 5. Conclusions

Altogether, these results suggest that FFW is a promising substrate for use as a feed to increase the lipid content in ZML and that solidified FFWS is a valuable feed for manufacturing ZML oil as a biodiesel feedstock.

## Figures and Tables

**Figure 1 animals-13-01958-f001:**
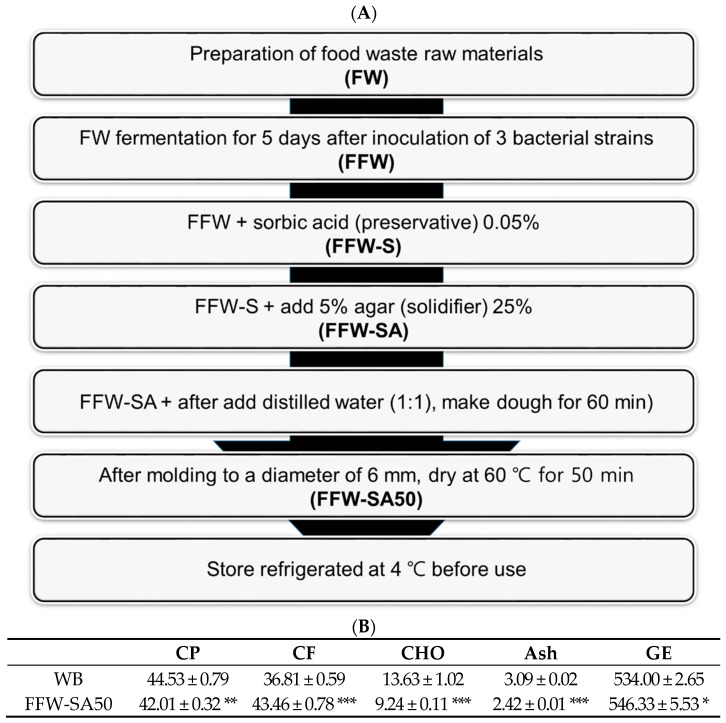
Manufacturing process of optimized fermented feed for increasing the fat content in ZML and analyses of the composition of ZML bred with the manufactured feed. (**A**) Manufacturing process of optimized feed for increasing the lipid content of ZML. The optimized feed was named FFW-SA50. (**B**) The composition of ZML bred with optimized feed. The values are presented as means ± SD. *p*-values < 0.05, 0.01, and 0.001 are indicated by *, **, and ***, respectively.

**Table 1 animals-13-01958-t001:** The composition of 10 single ingredients.

	CP (g/100 g)	CF (g/100 g)	CHO (g/100 g)	Ash (g/100 g)	GE (Kcal/100 g)
RB	15.7 ± 0.2 ^ef^	21.8 ± 0.1 ^a^	53.0 ± 0.1 ^c^	9.5 ± 0.0 ^b^	471.4 ± 0.5 ^a^
WB	15.0 ± 0.0 ^f^	4.0 ± 0.2 ^d^	76.2 ± 0.3 ^a^	4.9 ± 0.1 ^f^	399.8 ± 0.9 ^cd^
CS	8.3 ± 1.4 ^g^	3.4 ± 0.1 ^d^	78.6 ± 4.3 ^a^	6.5 ± 0.2 ^d^	378.1 ± 12.4 ^de^
SM	68.3 ± 2.0 ^a^	2.7 ± 0.1 ^e^	21.2 ± 1.9 ^e^	7.6 ± 0.2 ^c^	383.8 ± 1.8 ^cde^
SO	51.8 ± 2.5 ^b^	10.0 ± 0.3 ^c^	26.5 ± 1.9 ^e^	11.7 ± 0.3 ^a^	403.7 ± 0.2 ^c^
PO	44.4 ± 1.1 ^c^	10.4 ± 0.2 ^c^	39.3 ± 1.3 ^d^	5.9 ± 0.1 ^e^	428.4 ± 1.1 ^b^
SC	31.2 ± 1.8 ^d^	4.0 ± 0.1 ^d^	61.0 ± 1.7 ^b^	3.8 ± 0.1 ^g^	404.5 ± 0.4 ^c^
CG	16.6 ± 0.6 ^ef^	16.9 ± 0.1 ^b^	65.1 ± 0.6 ^b^	1.4 ± 0.0 ^h^	478.7 ± 0.5 ^a^
TF	20.1 ± 0.2 ^e^	1.1 ± 0.1 ^f^	68.1 ± 3.5 ^b^	6.6 ± 0.4 ^d^	362.2 ± 15.8 ^e^
FW	16.5 ± 1.9 ^ef^	8.1 ± 0.4 ^c^	61.4 ± 1.3 ^b^	8.0 ± 0.1 ^c^	384.7 ± 0.9 ^cde^

RB: rice bran; WB: wheat bran; CS: corn stalk; SM: defatted soybean meal; SO: defatted sesame meal; PO: defatted perilla meal; SC: soybean residue curd meal; CG: coffee grounds; TF: *Tenebrio molitor* fecal soil; FW: food waste; CP: crude protein; CF: crude fat; CHO: carbohydrate; GE: gross energy. Means denoted with the same superscript letters indicate no significant differences (*p* < 0.05), those marked with the different ones indicate significant differences (*p* < 0.05).

**Table 2 animals-13-01958-t002:** Weight gain (mg/ind.) of ZML bred with 10 single ingredients.

	5 Days	10 Days	15 Days	20 Days
RB	28.33 ± 4.45 ^b^	41.45 ± 14.86 ^bc^	71.31 ± 41.01 ^bc^	101.57 ± 36.61 ^de^
WB	35.80 ± 19.08 ^ab^	61.47 ± 29.40 ^abc^	112.17 ± 38.48 ^ab^	166.46 ± 43.16 ^a^
CS	24.67 ± 9.22 ^abc^	55.53 ± 29.59 ^bc^	69.44 ± 30.42 ^b^	99.75 ± 29.87 ^de^
SM	24.53 ± 11.32 ^a^	65.41 ± 39.38 ^bc^	93.22 ± 39.96 ^b^	139.78 ± 2.07 ^ab^
SO	20.13± 12.70 ^c^	68.73 ± 57.08 ^abcd^	79.01 ± 56.94 ^bc^	108.77 ± 30.68 ^cd^
PO	21.47 ± 11.57 ^c^	60.64 ± 25.58 ^bc^	898.25 ± 72.03 ^abc^	143.83 ± 69.21 ^ab^
SC	28.73 ± 17.72 ^abc^	46.36 ± 19.89 ^cd^	96.85 ± 38.48 ^ab^	124.92 ± 32.86 ^c^
CG	15.20 ± 9.62 ^d^	38.40 ± 27.90 ^cd^	76.24 ± 29.78 ^b^	103.32 ± 53.48 ^de^
TF	26.93 ± 19.43 ^bc^	58.80 ± 33.55 ^abc^	96.60 ± 37.96 ^ab^	132.58 ± 29.16 ^b^
FW	29.67 ± 29.62 ^abcd^	63.39 ± 36.27 ^abc^	104.48 ± 34.81 ^ab^	159.90 ± 62.85 ^ab^

RB: rice bran; WB: wheat bran; CS: corn stalk; SM: defatted soybean meal; SO: defatted sesame meal; PO: defatted perilla meal; SC: soybean residue curd meal; CG: coffee grounds; TF: *Tenebrio molitor* fecal soil; FW: food waste. Means denoted with the same superscript letters indicate no significant differences (*p* < 0.05), those marked with the different ones indicate significant differences (*p* < 0.05).

**Table 3 animals-13-01958-t003:** The composition of ZML bred with 10 single ingredients.

	CP (g/100 g)	CF (g/100 g)	CHO (g/100 g)	Ash (g/100 g)	GE (Kcal/100 g)
RB	43.7 ± 0.6 ^d^	40.1 ± 0.6 ^bc^	11.4 ± 0.2 ^abc^	2.9 ± 0.0 ^bc^	581.7 ± 2.1 ^ab^
WB	44.5 ± 0.8 ^cd^	36.8 ± 0.6 ^cd^	13.6 ± 1.0 ^a^	3.1 ± 0.0 ^a^	534.0 ± 2.7 ^e^
CS	48.3 ± 0.4 ^ab^	38.5 ± 2.6 ^cd^	7.8 ± 2.2 ^c^	3.2 ± 0.0 ^a^	573.3 ± 12.5 ^bc^
SM	45.2 ± 1.3 ^bcd^	42.9 ± 0.8 ^ab^	7.5 ± 1.9 ^c^	2.7 ± 0.0 ^d^	597.3 ± 3.5 ^a^
SO	46.8 ± 1.0 ^bcd^	37.1 ± 1.1 ^cd^	11.7 ± 1.2 ^ab^	3.1 ± 0.0 ^a^	568.0 ± 5.6 ^bc^
PO	48.6 ± 0.5 ^ab^	35.6 ± 0.3 ^d^	11.1 ± 0.6 ^abc^	2.8 ± 0.0 ^cd^	558.6 ± 1.5 ^cd^
SC	47.1 ± 1.1 ^abc^	38.9 ± 1.4 ^cd^	8.8 ± 0.3 ^bc^	3.0 ± 0.0 ^ab^	573.3 ± 7.2 ^bc^
CG	50.1 ± 0.3 ^a^	35.4 ± 0.4 ^d^	8.9 ± 0.1 ^bc^	3.1 ± 0.0 ^a^	555.0 ± 2.0 ^cd^
TF	47.2 ± 0.8 ^abc^	37.1 ± 0.8 ^cd^	1017 ± 1.2 ^abc^	3.2 ± 0.0 ^a^	562.3 ± 4.0 ^bcd^
FW	39.0 ± 0.7 ^e^	46.3 ± 0.4 ^a^	9.7 ± 0.4 ^bc^	2.7 ± 0.1 ^cd^	548.3 ± 6.0 ^de^

RB: rice bran; WB: wheat bran; CS: corn stalk; SM: defatted soybean meal; SO: defatted sesame meal; PO: defatted perilla meal; SC: soybean residue curd meal; CG: coffee grounds; TF: *Tenebrio molitor* fecal soil; FW: food waste. Means denoted with the same superscript letters indicate no significant differences (*p* < 0.05), those marked with the different ones indicate significant differences (*p* < 0.05).

**Table 4 animals-13-01958-t004:** Analyses of fermentation capacities of six microbial strains.

	Amylase (U/mL)	Maltase (U/mg Protein)	Sucrase (U/mL)	Protease (%)	Lipase (U/mL)
*L. plantarum*	3.47 ± 0.24 ^b^	35.28 ± 3.84 ^ca^	44.09 ± 3.35 ^ba^	107.70 ± 0.48 ^c^	0.086 ± 0.001 ^b^
*L. fermentum*	4.01 ± 0.50 ^a^	85.13 ± 6.89 ^aa^	24.94 ± 1.45 ^ca^	112.35 ± 0.82 ^b^	0.094 ± 0.002 ^a^
*L. acidophilus*	3.56 ± 0.12 ^b^	15.58 ± 3.56 ^da^	73.96 ± 0.12 ^aa^	116.87 ± 0.98 ^a^	0.093 ± 0.001 ^a^
*P. acidilactici*	2.77 ± 0.29 ^c^	28.98 ± 7.34 ^bc^	43.24 ± 3.52 ^ba^	112.59 ± 0.74 ^b^	0.093 ± 0.001 ^a^
*B. subtilis*	3.11 ± 0.09 ^b^	25.27 ± 1.01 ^ca^	46.31 ± 0.24 ^ba^	100.00 ± 0.00 ^e^	0.090 ± 0.001 ^a^
*S. cerevisiae*	3.34 ± 0.12 ^b^	42.68 ± 2.64 ^ba^	42.70 ± 0.80 ^bc^	106.85 ± 0.57 ^d^	0.086 ± 0.002 ^b^

Means denoted with the same superscript letters indicate no significant differences (*p* < 0.05), those marked with the different ones indicate significant differences (*p* < 0.05).

**Table 5 animals-13-01958-t005:** Body weight gain (mg/ind.) of ZML fed with fermented FW.

	5 Days	10 Days	15 Days	20 Days
(A) Body weight gain (mg/ind.) of ZML fed with EM-fermented FW				
FW	51.00 ± 6.00 ^c^	61.33 ± 6.03 ^d^	74.33 ± 13.32 ^c^	110.00 ± 11.14 ^c^
FW/EM/5 DAF	80.33 ± 11.85 ^b^	94.00 ± 31.61 ^ab^	74.00 ± 32.74 ^ab^	108.00 ± 39.15 ^bc^
FW/EM/7 DAF	77.26 ± 15.51 ^b^	92.30 ± 13.61 ^bc^	103.41 ± 14.70 ^bc^	118.41 ± 14.50 ^ab^
FW/EM/14 DAF	89*.26 ± 15.45 ^b^	54.56 ± 11.95 ^c^	84.74 ± 17.60 ^bc^	110.02 ± 24.47 ^ab^
FW/EM/21 DAF	112.33 ± 40.43 ^a^	95.96 ± 6.23 ^b^	74.19 ±8.98 ^b^	113.96 ± 9.23 ^a^
FW/EM/28 DAF	86.33 ± 3.21 ^ab^	92.00 ± 19.00 ^a^	93.00 ±17. 69 ^a^	122.67± 7.09 ^a^
(B) Body weight gain (mg/ind.) of ZML fed with yeast (S. cerevisiae)-fermented FW				
FW	51.00 ± 6.00 ^b^	61.33 ± 6.03 ^d^	74.33 ± 13.32 ^c^	110.00 ± 11.14 ^bc^
FW/Y/5 DAF	16.00 ± 5.00 ^c^	84.67 ± 13.28 ^ab^	67.33 ± 9.87 ^ab^	102.00 ± 7.81 ^ab^
FW/Y/7 DAF	31.00 ± 10.58 ^b^	82.75 ±2.82 ^b^	68.75 ± 11.10 ^b^	110.92 ± 15.92 ^a^
FW/Y/14 DAF	54.42 ± 2.92 ^a^	92.17 ± 8.81 ^a^	80.08 ± 19.81 ^ab^	106.83 ± 11.30 ^a^
FW/Y/21 DAF	33.33 ± 10.79 ^b^	77.67 ± 2.31 ^b^	70.33 ± 7.77 ^a^	98.33 ± 5.03 ^b^
FW/Y/28 DAF	21.48 ± 6.24 ^c^	71.33 ± 5.77 ^c^	56.50 ± 19.81 ^bc^	96.75 ± 2.17 ^b^
(C) Body weight gain (mg/ind.) of ZML fed with 3M-fermented FW				
FW	51.00 ± 6.00 ^a^	61.33 ± 6.03 ^c^	74.33 ± 13.32 ^bc^	110.00 ± 11.14 ^c^
FW/3M/5 DAF	32.26 ± 18.92 ^ab^	85.41 ± 19.00 ^ab^	73.26 ± 24.88 ^ab^	129.96 ± 15.35 ^a^
FW/3M/7 DAF	16.67 ± 10.26 ^bc^	78.67 ± 16.26 ^a^	70.00 ± 8.54 ^a^	124.33 ± 13.65 ^a^
FW/3M/14 DAF	18.33 ± 16.29 ^bc^	79.41 ± 19.52 ^ab^	59.37 ± 18.32 ^bc^	119.41 ± 13.65 ^ab^
FW/3M/21 DAF	38.74 ± 43.95 ^abc^	80.52 ± 16.28 ^b^	62.56 ± 10.08 ^b^	128.07 ± 8.06 ^b^
FW/3M/28 DAF	30.00 ± 16.82 ^ab^	74.67 ± 7.23 ^ab^	65.00 ± 8.54 ^b^	125.67 ± 23.71 ^ab^

FW, food waste; EM, effective microorganisms; Y, yeast; 3M, selected three microorganisms; DAF, days after fermentation. Means denoted with the same superscript letters indicate no significant differences (*p* < 0.05), those marked with the different ones indicate significant differences (*p* < 0.05).

**Table 6 animals-13-01958-t006:** Body weight gain (mg/ind.) of ZML fed with FFW containing sorbic acid or grapefruit seed extract.

	5 Days	10 Days	15 Days	20 Days
FW	51.00 ± 6.00 ^d^	61.33 ± 6.03 ^c^	74.33 ± 13.32 ^d^	110.00 ± 11.14 ^b^
FFW/S 0.05%	86.67 ± 24.50 ^a^	111.48 ± 9.96 ^ab^	93.50 ± 24.71 ^bc^	165.67 ± 16.44 ^a^
FFW/S 0.10%	73.33 ± 11.93 ^b^	142.33 ± 3.79 ^a^	128.00 ± 8.198 ^a^	166.67 ± 6.11 ^a^
FFW/S 0.15%	61 ± 10.00 ^c^	107.33 ± 9.61 ^b^	107.33 ± 23.69 ^ab^	171.33 ± 13.61 ^a^
FFW/GSE 0.05%	62.67 ± 20.13 ^b^	106.37 ± 30.80 ^ab^	73.85 ± 9.77 ^c^	161.33 ± 13.50 ^a^
FFW/GSE 0.10%	62.66 ± 8.32 ^bc^	102.04 ± 16.47 ^b^	111.52 ± 27.98 ^ab^	147.85 ± 18.07 ^ab^
FFW/GSE 0.15%	54.00 ± 23.07 ^bc^	85.33 ± 15.70 ^b^	92.67 ± 12.50 ^c^	136.00 ± 11.14 ^b^

FFW, food waste fermented at 3M for 5 days; S, sorbic acid; GSE, grapefruit seed extract. Means denoted with the same superscript letters indicate no significant differences (*p* < 0.05), those marked with the different ones indicate significant differences (*p* < 0.05).

**Table 7 animals-13-01958-t007:** Body weight gain (mg/ind.) of ZML fed with solidified fermented feed by agar, carrageenan, or starch.

	5 Days	10 Days	15 Days	20 Days
WB	70.00 ± 30.00 ^bc^	110.00 ± 30.00 ^b^	136.67 ± 11.55 ^b^	150.00 ± 10.00 ^c^
FW	50.00 ± 34.64 ^cd^	110.00 ± 26.46 ^b^	133.33 ± 25.17 ^b^	153.33 ± 15.28 ^c^
FFWS	73.33 ± 23.09 ^b^	123.33 ± 30.55 ^ab^	166.67 ± 20.82 ^ab^	203.33 ± 11.55 ^ab^
FFWS/A/50	50.00 ±30.00 ^cd^	126.67 ± 20.82 ^b^	166.67 ± 15.28 ^a^	230.00 ± 10.00 ^a^
FFWS/A/100	13.33 ± 5.77 ^d^	103.33 ± 15.28 ^a^	150.00 ± 20.00 ^b^	210.00 ± 10.00 ^b^
FFWS/A/150	80.00 ± 52.92 ^ab^	106.67 ± 49.33 ^ab^	136.67 ± 49.33 ^b^	223.33 ± 37.86 ^b^
FFWS/C/50	43.33 ± 15.28 ^d^	130.00 ± 36.06 ^ab^	173.33 ± 20.82 ^a^	220.00 ± 26.46 ^ab^
FFWS/C/100	66.67 ± 41.63 ^bc^	113.33 ± 11.55 ^b^	160.00 ± 36.06 ^ab^	193.33 ± 37.86 ^ab^
FFWS/C/150	60.00 ± 45.83 ^bcd^	113.33 ± 23.09 ^b^	143.33 ± 28.87 ^b^	186.67 ± 15.28 ^c^
FFWS/S/50	63.33 ± 37.86 ^bcd^	126.67 ± 25.17 ^a^	176.67 ± 46.19 ^ab^	210.00 ± 36.06 ^ab^
FFWS/S/100	86.67 ± 5.77 ^a^	113.33 ± 5.77 ^b^	150.00 ± 51.96 ^b^	196.67 ± 28.87 ^bc^
FFWS/S/150	70.00 ± 34.64 ^ab^	106.67 ± 30.55 ^ab^	136.67 ± 15.28 ^ab^	163.33 ± 5.77 ^bc^

FFWS, fermented food waste containing 0.15% sorbic acid; A, agar; C, carrageenan; S, starch. Means denoted with the same superscript letters indicate no significant differences (*p* < 0.05), those marked with the different ones indicate significant differences (*p* < 0.05).

## Data Availability

Not applicable.

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
