# Peer review of "Development of Optimized Feed for Lipid Gain in Zophobas morio (Coleoptera: Tenebrionidae) Larvae"

_animals, 2023, doi:10.3390/ani13121958_

Round 1
Reviewer 1 Report
So, an interesting piece of work but the paper needs a) to be edited by someone who is more familiar with English since there are multiple grammar issues throughout the manuscript. I’ve highlight some of them on page 2 but there are many others.
See my comments regarding tables 1,2, 3 and 5. In particular expressing weight gain in % rather than showing actual larval weights is problematic.
Page |
Line |
Comments |
2 |
50 |
So is the 16.7-57.9% on a dry matter basis (likely) or as is. |
2 |
51 |
Change “monitored” to “modified” |
2 |
61 |
Presumably this is on a dry matter basis but it should be stated. |
2 |
63 |
Delete “powerful” |
2 |
68 |
What is fecal soil. Do you mean frass? Also, the authors feed to better define what is in the “food waste” |
2 |
76 |
Delete “to” |
2 |
85 |
What is Caloriesin? Presumably the authors mean energy as calories are a unit of measure. |
4 & 5 |
Tables 1 & 3 |
Both these tables show result son analysis of ingredients and larvae but do the authors really the results should be expressed to 2 decimal places. That infers of level of accuracy which is unlikely in this reviewer’s opinion. |
4 & 6 |
Table 2 & 5 |
What is weight gain being express as a percentage. That’s potentially can be very misleading. Far better to express the actual larval weights (in mg) to better allow the reader to evaluate these data. |
/
Author Response
We sincerely appreciate your insightful comments. We have carefully considered your feedback and would like to provide the following response. We hope our response serves as appropriate answers to your feedback.
So, an interesting piece of work but the paper needs a) to be edited by someone who is more familiar with English since there are multiple grammar issues throughout the manuscript. I’ve highlight some of them on page 2 but there are many others.
See my comments regarding tables 1,2, 3 and 5. In particular expressing weight gain in % rather than showing actual larval weights is problematic.
Page |
Line |
Comments |
2 |
50 |
So is the 16.7-57.9% on a dry matter basis (likely) or Reply : It is a dry matter basis. Therefore, we changed “approximately 16.7%-57.9%” to “approximately 16.7%-57.9% of total dried body weight”. |
2 |
51 |
Change “monitored” to “modified” Reply : We changed “monitored” to ‘modified’. |
2 |
61 |
Presumably this is on a dry matter basis but it should be stated. Reply : We have added “dried” in front of ZML |
2 |
63 |
Delete “powerful” Reply : We deleted |
2 |
68 |
What is fecal soil. Do you mean frass? Also, the authors feed to better define what is in the “food waste” Reply : Yes. Fecal soil is also referred as frass. We used "feed-conversed food waste" as "food waste". Therefore, we corrected “food waste” to “feed-conversed food waste”. |
2 |
76 |
Delete “to” Reply : We deleted |
2 |
85 |
What is Caloriesin? Presumably the authors mean energy as calories are a unit of measure. Reply : Caloriesin is miswritten. We missed spacing between "Calories" and "in". So, we corrected “Caloriesin” to “Calories in”. |
4 & 5 |
Tables 1 & 3 |
Both these tables show results on analysis of ingredients and larvae but do the authors really the results should be expressed to 2 decimal places. That infers of level of accuracy which is unlikely in this reviewer’s opinion. Reply : We have expressed them to 1 decimal place according to your suggestion. |
4 & 6 |
Table 2 & 5 |
What is weight gain being express as a percentage. That’s potentially can be very misleading. Far better to express the actual larval weights (in mg) to better allow the reader to evaluate these data. Reply : We changed the percentage to the actual larval weights as mg/ind. |
Reviewer 2 Report
Interesting study; how did you decide with which microbes to initiate the fermentation process?
Really minor edit - line 51 - suggest replacing the word "monitored" with "modified" or "altered"
Line 123 and also within lines 233-240 - did you reanalyze diets when they were complete with the agar, carageenen, and/or starch added?; suggest you provide composition of the final total diet composition that resulted in the body compositions reported.
In references cited, numbers 18 and 37 refer to the same publication. The Zoo Biology references within the same citations list do not require the verbiage "in association with the American Association of Zoos and Aquarium" - this is a legitimate scientific journal and Zoo Biology (volume, issue, and page numbers) alll that are needed.
Author Response
We sincerely appreciate your insightful comments. We have carefully considered your feedback and would like to provide the following response. We hope our response serves as appropriate answers to your feedback.
Interesting study; how did you decide with which microbes to initiate the fermentation process?
Reply
: Thanks for your comment. We selected previously reported microbes. The references were already mentioned in Result 3.3.
Really minor edit - line 51 - suggest replacing the word "monitored" with "modified" or "altered"
Reply
: Thanks for your comment. We replaced the word “monitored” to “modified”.
Line 123 and also within lines 233-240 - did you reanalyze diets when they were complete with the agar, carageenen, and/or starch added?; suggest you provide composition of the final total diet composition that resulted in the body compositions reported.
Reply
: Thanks for your comment. We analyzed the composition of diets containing solidifying materials and found that the composition of each diet was similar to one another. Therefore, we did not show the information in manuscript. The data was presented as supplementary information.
In references cited, numbers 18 and 37 refer to the same publication. The Zoo Biology references within the same citations list do not require the verbiage "in association with the American Association of Zoos and Aquarium" - this is a legitimate scientific journal and Zoo Biology (volume, issue, and page numbers) all that are needed.
Reply
: Thanks for your comment. We corrected it. The reference was inserted as #20.